# Structural Transitions of Alpha-Amylase Treated with Pulsed Electric Fields: Effect of Coexisting Carrageenan

**DOI:** 10.3390/foods11244112

**Published:** 2022-12-19

**Authors:** Junzhu Li, Jiayu Zhang, Chen Li, Wenjing Huang, Cheng Guo, Weiping Jin, Wangyang Shen

**Affiliations:** 1School of Food Science and Engineering, Wuhan Polytechnic University, Wuhan 430023, China; 2Hubei Key Laboratory for Processing and Transformation of Agricultural Products, Wuhan 430023, China; 3Key Laboratory for Deep Processing of Major Grain and Oil, Ministry of Education, Wuhan 430023, China

**Keywords:** molten globule, protein, pulsed electric field, λ-carrageenan, assembly behavior

## Abstract

Pulsed electric field (PEF) is an effective way to modulate the structure and activity of enzymes; however, the dynamic changes in enzyme structure during this process, especially the intermediate state, remain unclear. In this study, the molten globule (MG) state of α-amylase under PEF processing was investigated using intrinsic fluorescence, surface hydrophobicity, circular dichroism, etc. Meanwhile, the influence of coexisting carrageenan on the structural transition of α-amylase during PEF processing was evaluated. When the electric field strength was 20 kV/cm, α-amylase showed the unique characteristics of an MG state, which retained the secondary structure, changed the tertiary structure, and increased surface hydrophobicity (from 240 to 640). The addition of carrageenan effectively protected the enzyme activity of α-amylase during PEF treatment. When the mixed ratio of α-amylase to carrageenan was 10:1, they formed electrostatic complexes with a size of ~20 nm, and carrageenan inhibited the increase in surface hydrophobicity (<600) and aggregation (<40 nm) of α-amylase after five cycles of PEF treatment. This work clarifies the influence of co-existing polysaccharides on the intermediate state of proteins during PEF treatment and provides a strategy to modulate protein structure by adding polysaccharides during food processing.

## 1. Introduction

Alpha-amylase (EC 3.2.1.1) is widely used in brewing, baking, and juice clarification [1]. The enzyme activity of α-amylase is sensitive to the environment because it belongs to a protein. Small specific changes in environmental conditions will induce a conformational transition of the protein from its native to its unfolded state. This transition is not a one-step reaction; it comprises a metastable state between the native and the unfolded state called the “molten globule” (MG) [2]. The presence of an MG state has been reported in food processing, such as in the acid-induced unfolding of bovine serum albumin [3] and soybean 11S globulin [4], the semicarbazide-induced denaturation of α-lactoalbumin [5], and sorbitol-induced myoglobulin aggregation [6], etc. Generally, the transition of the protein from its native to its MG state is reversible, but progression from the MG state to the denatured or aggregated state is irreversible [7]. Furthermore, the MG state is crucial in determining the protein folding pathways and influencing the activity and functional properties of the protein [8,9].

The MG state of α-amylase with partial unfolding at an acidic pH (3.0) was reported by Asghari et al. [10]. Then, Shokri et al. found that both the thermophilic and mesophilic α-amylase showed the characteristics of an MG state at pH 4.0 [11]. They also reported that co-existing calcium ions, trehalose and sorbitol had a positive effect on the refolding of mesophilic α-amylase from an MG state to its native state. Therefore, the co-existing components are of great interest in the structural transition of a protein during food processing. Polysaccharides can significantly affect the structure of a protein mainly through electrostatic interactions and hydrogen bonding [12]. For example, Yurij et al. found that λ-carrageenan increased the helix of lysozymes, exposed tryptophan residues to a hydrophobic nonpolar environment, and decreased the thermal stability of lysozymes [13]. Wu et al. found that an acidic branched polysaccharide from green tea could bind with α-amylase and inhibit enzyme activity in a low concentration range (0.79–7.88 μmol L^−1^) [14]. However, the effect of co-existing polysaccharides on the unfolding of proteins during certain food processing is limited. 

Pulsed electric field (PEF) is a non-thermal processing method for protein modification [15]. The electric field disturbs the charge density of amino acids at the -COOH and -NH_3_^+^ moieties, resulting in the polarization of the peptide bonds [16,17]. The conformation, solubility, and surface hydrophobicity of the proteins are correspondingly changed after PEF treatment. Wang et al. reported that moderate PEF treatment led to the partial structural unfolding of a myofibrillar protein, thereby exposing more interior hydrophobic and sulfhydryl groups on the surface of the protein [18]. Wei et al. found that an α-helix was more sensitive to PEF treatment, and the unfolding of ovalbumin reached the maximum when it was treated with a 27 kV electric field for 4320 μs [19]. The papain lost 64% of its activity after it was subjected to 13 kV/cm and 288 pulses of PEF treatment. Meanwhile, the SH groups in the papain decreased to 80% [15]. Molecular dynamic simulations of PEF-treated trypsin demonstrated that PEF activated enzymes through an increase in the number of molecular hydrogen bonds and the solvent-accessible surface area [20]. Current research is focused on the final structure, activity, and functional properties of proteins treated with PEF, but the study of the intermediate state is neglected. 

The regulation of enzyme structures and the activity of α-amylase is valuable in food processing [21]. In this work, the molten globule (MG) state of α-amylase under moderate PEF treatment was investigated using intrinsic fluorescence, surface hydrophobicity, circular dichroism, etc. Meanwhile, the influence of coexisting carrageenan on the structural transition of α-amylase during PEF processing was evaluated via enzyme activity, particle size, and sodium dodecyl sulfate-polyacrylamide gel electrophoresis (SDS-PAGE). This study aims to provide a theoretical basis for understanding the effect of coexisting polysaccharides on the unfolding process of α-amylase induced by PEF treatment. 

## 2. Materials and Methods

### 2.1. Materials

Alpha-amylase (EC 3.2.1.1) originated from *Bacillus licheniformis*, λ-carrageenan, and 8-anilino-1-naphthalenesulfonic acid ammonium salt (8-ANS) were purchased from Aladdin Biochemical Technology Co., Ltd. (Shanghai, China). Other chemicals, including NaCl, soluble starch, and Coomassie brilliant blue, were acquired from Sinopharm Chemical Reagent Co., Ltd. (Shanghai, China). 

### 2.2. Sample Preparation

Alpha-amylase was dissolved in 10 mM NaCl at a concentration of 0.2 mg/mL. The λ-carrageenan powder was suspended in 10 mM NaCl, stirred at 80 °C for 2 h, and cooled to room temperature. All stock solutions were placed in a refrigerator at 4 °C overnight to fully hydrate. The α-amylase and λ-carrageenan stock solutions were mixed at ratios of 1:1, 10:1, and 100:1, whereas the fixed concentration of α-amylase was 0.2 mg/mL, and the final pH was set to 5.0. The conductivity of the mixtures was approximately 1.4 mS/cm, which was measured at 25 °C using a conductivity meter (SX650, Sanxin, Shanghai, China).

### 2.3. Pulsed Electric Field (PEF) Treatment

The experiment was carried out using a THU-PEF4 continuous PEF device (New prospect laboratory systems engineering integrator, Wuhan, China). The PEF instrument consists of three major parts: the liquid processing chamber, a peristaltic pump, and a pulsed electric field generator. The diameter of the parallel-plate electrode is 0.3 cm, and the gap distance is 0.3 cm. A digital oscilloscope (TEKWAY DST1102B, 100 MHz, 1 G Sa/s, Hangzhou, China) was used to monitor a pulse width of 40 μs and a frequency of 1.06 kHz. Two thermometers were placed at the inlet and outlet of the processing chamber to record the temperature during PEF processing. The flow rate of the liquid was set at 100 mL/min, and the electric field strength (*E*) ranged from 0 to 20 kV/cm.

### 2.4. Measurement of Enzyme Activity

Enzyme activity was measured following the method described in Jin’s study [22]. Briefly, 40 µL of soluble starch solution (2.0 g/L) and 40 µL of samples were mixed in a microplate. The covered microplate was incubated at 50 °C for 30 min, and 20 µL of 1 M HCl was added to stop the reaction. Next, 100 µL of iodine reagent (5 mM I_2_ and 25 mM KI) was bound to the unreacted starch. Finally, 150 µL of the reactive solution was removed to measure the absorbance at 580 nm. Enzyme activity was calculated according to the standard curve (y = 00273x, *r*^2^ = 0.99).
U/mL = (A_580 control_ − A_580 sample_) ÷ A_580_/mg starch ÷ 30 min ÷ 0.04 mL
where A_580 control_ is the absorbance obtained from the starch without the addition of the enzyme, A_580 sample_ is the absorbance of the starch digested with the enzyme, A_580_/mg starch is the absorbance for 1 mg of starch as derived from the standard curve, 30 min is the incubation time, and 0.04 mL is the volume of the enzyme.

### 2.5. Intrinsic Fluorescence

The intrinsic fluorescence of the α-amylase and the mixtures was measured via a fluorescence spectrophotometer (F-4600, HITACHI, Tokyo, Japan) with an excitation wavelength of 280 nm and emission spectrum of 300–500 nm. The scan speed was set at 240 nm/min.

### 2.6. Circular Dichroism (CD) Studies

The secondary structure of the α-amylase and the mixtures was investigated using spectroscopy (JASCO J-1500, Tokyo, Japan). The concentration of α-amylase was 0.2 mg/mL, and the CD spectrum was recorded in the range of 190 to 240 nm using 0.1 cm quartz cells with a scan speed of 100 nm/min [23].

### 2.7. Surface Hydrophobicity (S_0_)

The α-amylase and the mixtures were diluted 20, 40, 50, 80 and 100 times, and each sample (10 µL) was mixed with 8-ANS of stock solution (40 µL, 8 mM). The spectrum was measured using a fluorescence spectrophotometer at an excitation wavelength of 380 nm and an emission wavelength of 470 nm. Linear regression analysis was used to calculate the initial slope of the fluorescence intensity versus the enzyme concentration, which was calculated as *S_0_* [24]. 

### 2.8. Dynamic Light Scattering and Surface Charges

The particle size distribution and zeta potential of the α-amylase samples after PEF treatment were measured via a Malvern ZS 90 instrument (Malvern Instrument, Ltd., Worcestershire, UK).

### 2.9. Sodium Dodecyl Sulfate-Polyacrylamide Gel Electrophoresis (SDS-PAGE)

The experiment then followed the protocol of the SDS-PAGE commercial kit (Beyotime Biotechnology, Shanghai, China). The α-amylase samples (0.2 mg/mL, 40 μL) were mixed with 10 μL of loading buffer and then boiled for 5 min. Electrophoresis was run at 60 V for 30 min in the stacking gel (5%) and at 120 V for 70 min in the separating gel (12%). After electrophoresis, the gel was stained using Coomassie brilliant blue R-250 for 30 min and then decolorized. The gel was imaged using the Bio-Rad imaging system (Bio-Rad Laboratories, Hercules, CA, USA).

### 2.10. Statistics Analyses

Unless otherwise specified, all experiments were performed at least in triplicate, and the results, shown in figures are represented as mean ± standard deviation. Statistical analysis was performed via a one-way ANOVA using SPSS 23 (SPSS Inc.; Chicago, IL, USA) to evaluate significance (*p* < 0.05).

## 3. Results and Discussion

### 3.1. MG State of α-Amylase Induced by PEF during Unfolding Process

The influence of the field strength (*E*, 0 to 20 kV/cm) on the enzyme activity of α-amylase is shown in Figure 1a. In the low *E* region (<10 kV/cm), enzyme activity remained at a high level, but when *E* enhanced to 20 kV/cm, they significantly decreased to 92%. Tryptophan (Try) is commonly used as an indicator for detecting the conformation of a protein [25] because it is sensitive to the microenvironment. The sequence of α-amylase contains 17 Try residues, and the maximum intensity of its intrinsic fluorescence spectra was located at ~350 nm. After PEF treatment, the intrinsic fluorescence intensity of α-amylase gradually decreased (Figure 1b). The ratio of fluorescence intensity obtained at 330 and 350 nm could reflect small changes in the tertiary structure of α-amylase. A decrease in the *I*_330_/*I*_350_ ratio is indicative of a red shift, whereas an increase in this ratio signifies a blue shift. The ratio of *I*_330_/*I*_350_ had the lowest value (0.53) at 15 kV/cm, suggesting that the Try residues were exposed to the polar solvent. Similar quenching phenomena were also observed in the acid-induced denaturation of α-amylase, indicating the presence of a non-native stable intermediate [11]. When the quenching rate was limited to 20%, the influence of the field strength on enzyme activity was not significant (*p* < 0.05). It meant that the transformation of the α-amylase microstructure preceded changes in its activity. The CD spectrum showed that the initial secondary structure of α-amylase was 32.1% α-helix, 9.7% β-sheet, 27.6% β-turn, and 30.6% random coil after being fitted to the curve using Young’s modulus (Figure 1c). When *E* reached 20 kV/cm, the amount of α-helix (31.0%) and β-turn (26.0%) decreased, but the amount of β-sheet (11.8%) and random coil (31.1%) increased. However, the amplitude of change in the secondary structure of α-amylase was not significant after one cycle of PEF treatment, indicating that denatured α-amylase retained a pronounced secondary structure. Surface hydrophobicity (*S_0_*) is critical in understanding the unfolding/refolding of a protein [26,27]. The value of *S*_0_ increased dramatically when the *E* was 20 kV/cm (Figure 1d). The enhancement of ANS binding emphasized the existence of hydrophobic surface residues on the surface, which is an index of the molten globule state [28]. The characteristics of the MG state include (i) a substantial secondary structure, (ii) a loose tertiary structure without compact side-chain packing, and (iii) the presence of more hydrophobic residues on the surface. The results of intrinsic fluorescence, CD, and surface hydrophobicity for PEF treated α-amylase showed that an intermediate state existed before complete denaturation, that is, the MG state [28,29].

### 3.2. Effect of Carrageenan on the Unfolding of α-Amylase Induced by PEF

#### 3.2.1. Effect of Carrageenan on Enzyme Activity

The enzyme activities of α-amylase and the α-amylase/λ-carrageenan mixtures after PEF treatment are shown in Figure 2. When *E* was fixed at 20 kV/cm, the enzyme activity of α-amylase was effectively inhibited with an increase in PEF cycles. Furthermore, almost 50% of enzyme activity is lost after four cycles, suggesting the unfolding of α-amylase. Adding λ-carrageenan had a positive effect on maintaining the enzyme folding state and keeping a higher activity. When the ratio of α-amylase/λ-carrageenan was 1:1, enzyme activity was retained at approximately 100% after five cycles of PEF treatment. Interestingly, when the ratio of α-amylase/λ-carrageenan was 10:1, changes in enzyme activity were completely different from the other two ratios. A possible reason might be that when the λ-carrageenan was added to a certain level, it interacted with α-amylase and changed the intra-molecular or inter-molecular bindings. This phenomenon arouses interest, as shown in Figure 2.

#### 3.2.2. Effect of λ-Carrageenan on the Intrinsic Fluorescence

The intrinsic fluorescence reflects not solely the ternary structure of the protein but also suggests an interaction between the protein and other molecules [30]. The influence of carrageenan on the intrinsic fluorescence spectrum of α-amylase after PEF treatment is exhibited in Figure 3. First, λ-carrageenan decreased the intrinsic fluorescence of α-amylase without PEF treatment at all mixing ratios. This indicates that the tryptophan residues of α-amylase are moving to a less polar environment after binding with λ-carrageenan. The degree of quenching of the α-amylase/λ-carrageenan 10:1 mixture was the largest, suggesting they had a strong binding interaction [31]. After the second PEF cycle, the fluorescence intensity of α-amylase alone decreased dramatically, and the quenching rate reached more than 60%. The addition of λ-carrageenan did not change the quenching tendency, but the quenching rate decreased, indicating that carrageenan inhibited the conformational transition of α-amylase induced by PEF treatment.

#### 3.2.3. Effect of λ-Carrageenan on the Aggregation Behavior

The particle size distribution of the α-amylase and α-amylase/λ-carrageenan mixtures after PEF treatment is shown in Figure 4. The aggregation behavior induced by the PEF treatment of α-amylase was similar to that of egg and canola proteins [32,33]. Therefore, the inactivation of the enzyme first involves the unfolding of the native proteins and is followed by the irreversible aggregation of the unfolded molecules. Initially, α-amylase was sequentially assembled into a growing nucleus in a thermodynamically reversible manner. Then, the nucleus grew until a critical point (cycle two) and the aggregation of α-amylase became irreversible.

The original size of α-amylase was within 10 nm. After one cycle of PEF treatment, two peaks appeared in the size distribution, the smaller peak was 60–70 nm, and the larger peak was ~200 nm. After two–five cycles of PEF treatment, only particles with a size of 500–600 nm were observed. That means that the inactivation of α-amylase via PEF treatment usually involves the unfolding of the native states first (from 10 nm to ~60 nm) and is followed by the irreversible aggregation of the unfolded molecules (>500 nm). The particles with a size of approximately 60 nm might be attributed to the unfolded MG state, which is consistent with previous findings (Section 3.1). This intermediate state (MG) was also observed when λ-carrageenan was added in small amounts (with an α-amylase/λ-carrageenan ratio of 100:1). At this time, the carrageenan was insufficient to inhibit the structural transition of bulk α-amylase, but it delayed appearance of the MG state to the second cycle of PEF treatment. Furthermore, after five cycles of PEF treatments, the particle size of α-amylase/λ-carrageenan mixtures reached 530 nm, which was smaller than that of sole α-amylase (710 nm).

A peak was observed at 30 nm for each mixing ratio, presumably due to the formation of the α-amylase/λ-carrageenan complexes. The particle size of the α-amylase/λ-carrageenan mixtures (at ratios of 10:1 and 1:1) increased slowly with an increase in treated cycles of PEF treatment. Especially when the ratio was 10:1, the particle sizes of the mixtures were all less than 100 nm after five cycles of PEF treatment. Visually, there was also no obvious turbidity in the mixed solution. Because PEF treatment changed the dipole moment of the protein and induced the exposure of sulfhydryl and hydrophobic groups, the structure of the protein transformed from native to partial unfolding, resulting in protein aggregation [34]. 

#### 3.2.4. Formation of α-Amylase/λ-Carrageenan Complexes

Based on the results of fluorescence quenching and particle size distribution, it is speculated that α-amylase and λ-carrageenan could form electrostatic complexes in the current state. The ζ-potential values of α-amylase were sensitive to pH. At pH 6.0 and 7.0, the value of the ζ-potential was approximately −23 mV (Figure 5a). When the pH decreased to 5.0, the absolute value of the ζ-potential decreased significantly (*p* < 0.05), which suggests that there were some positively charged patches on the surface of α-amylase. These positively charged patches could interact with negatively charged groups in λ-carrageenan [22]. Meanwhile, the turbidity of the mixture is a direct indicator for evaluating the interaction. The α-amylase/λ-carrageenan (1:1) complexes had the lowest ζ-potential values, which were associated with excess λ-carrageenan (Figure 5b). As shown in Figure 5c, the turbidity first increased and then decreased with an increase in the α-amylase/λ-carrageenan ratio. Maximum turbidity was achieved at a ratio of 10:1, which implied that α-amylase/λ-carrageenan 10:1 was the optimal binding ratio. The autocorrelation function (g1) is obtained by capturing the Brownian motion of the particles, and it decays exponentially with a characteristic time [35]. In Figure 5d, α-amylase had a smaller particle size than the α-amylase/λ-carrageenan electrostatic complexes. When the ratios were 1:1 and 10:1, a second plateau was observed at 3000 μs and 1000 μs, respectively, suggesting the presence of two types of particles in the mixture. 

### 3.3. Proposed Mechanism of λ-Carrageenan Affecting the Unfolding of α-Amylase

#### 3.3.1. Secondary Structure of α-Amylase and Complexes after PEF Treatment

To explore the possible effects of λ-carrageenan on the unfolding process of α-amylase, the secondary structure, surface hydrophobicity, and aggregation of α-amylase and α-amylase/λ-carrageenan with a ratio of 10:1 were compared. The secondary structure of α-amylase and the complexes during PEF treatment is shown in Figure 6. The intensity of the negative peaks at 208 nm and 222 nm and the positive peak at 195–198 nm is ascribed to the α-helix and β-turn structure, respectively. After five cycles, the number of α-helices and β-turns in α-amylase decreased to 27.1% and 18.8%, respectively, while the percentage of β-sheets increased to 22% (nearly 2-fold). Based on the structural characteristics of α-amylase, the α-helices and β-turns played an important role in keeping catalysis [36]. As for the α-amylase/λ-carrageenan complexes (10:1), the initial secondary structure of α-amylase did not change. However, after PEF treatment, an increase in β-sheets and a decrease in α-helices and β-turns were observed. Similar changes in heat-treated α-amylase indicated denaturing conditions [37]. According to the molecular characteristics of α-amylase, the secondary structure of the α-helix played an important role in maintaining catalysis [36]. The binding between α-amylase and λ-carrageenan might have exposed the catalytic domain to the external electric field, which became less protective against enzyme inactivation.

#### 3.3.2. Surface Hydrophobicity of α-Amylase and Complexes after PEF Treatment

After five cycles of PEF treatment, the *S_0_* of α-amylase increased nearly 10-fold, implying that most hydrophobic groups inside the α-amylase were exposed (Figure 7a). In general, exposed hydrophobic regions on the protein surface promoted the intra-connections among molecules and induced the formation of large aggregates [28]. Similar phenomena were observed for ovalbumin and myofibrillar proteins after moderate PEF treatment [34,38]. In contrast, *S*_0_ did not increase dramatically after protein binding with polysaccharides (Figure 7b), reaching ~600 at cycle four. This phenomenon might be explained by the hydrophobic regions inside α-amylase bound to λ-carrageenan through electrostatic interactions, hydrogen bonds, or hydrophobic interactions. After PEF treatment, only a small amount of the hydrophobic region was exposed to combine with ANS, resulting in a lower *S*_0_ value. 

#### 3.3.3. SDS-PAGE of α-Amylase and Complexes after PEF Treatment

SDS-PAGE is a useful technique to study the aggregation behavior or degradation behaviors of proteins. As shown for the SDS-PAGE (Figure 8), after the second cycle of PEF treatment, the amount of soluble α-amylase decreased, and large aggregates appeared on the top of the separating gel. During PEF processing, protein molecules are first polarized, and then hydrophobic amino acids or sulfhydryl groups are exposed, which aggregate together to form insoluble proteins through hydrophobic interaction or covalent bonds [15]. With the further aggregation of insoluble proteins or an increase in particle size, the α-amylase was precipitated, and protein solubility was reduced. The addition of λ-carrageenan inhibited the exposure of the hydrophobic domains of α-amylase, reduced the aggregation behavior, and kept the solubility of α-amylase at a high level after five cycles. 

## 4. Conclusions

The structure, conformation transformation (folding or unfolding), and catalysis ability of α-amylase have a close relationship. The results of the intrinsic fluorescence, secondary structure, surface hydrophobicity, and aggregation behavior indicated that α-amylase exhibited a molten globular state after treatment with PEF at 20 kV/cm for one cycle. The maximum inhibition effect was observed after five cycles of PEF treatment. However, the addition of λ-carrageenan protected α-amylase from PEF-induced unfolding. When α-amylase was mixed with λ-carrageenan at a ratio of 10:1, nanoscale electrostatic complexes (~100 nm) were formed, which inhibited the increase in surface hydrophobicity and changed the secondary structure of α-amylase. However, the complexation also limited the binding between the enzyme and the substrate, resulting in poor protection of enzyme activity. This work clarifies the intermediate state of α-amylase during external electric field-induced unfolding. Additionally, it was found that polysaccharides play an important role in regulating this unfolding process, and it provides key parameters for controlling enzyme catalysis during PEF processing.

## Figures and Tables

**Figure 1 foods-11-04112-f001:**
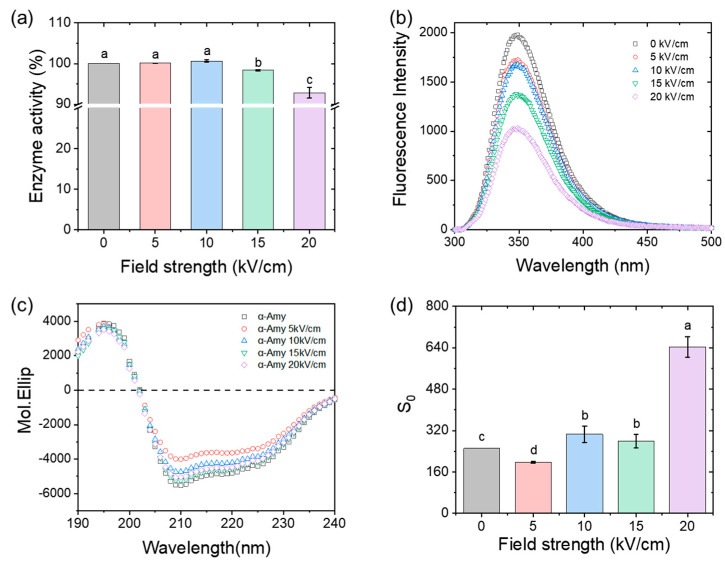
MG state of α-amylase induced by PEF (1 cycle). (**a**) Effect of electric field on enzyme activities, (**b**) intrinsic fluorescence intensity, (**c**) CD spectra, and (**d**) surface hydrophobicity. Letters a-d inserted in figures for degree of significance, *p* < 0.05.

**Figure 2 foods-11-04112-f002:**
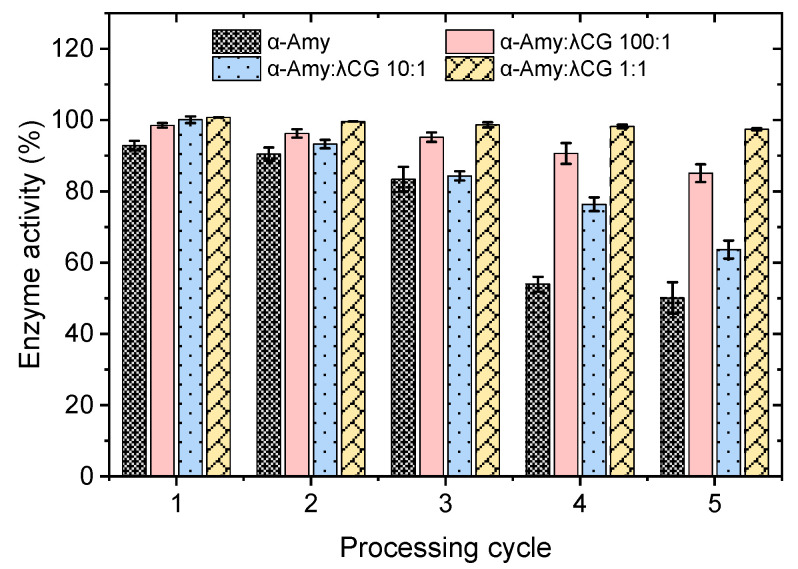
Enzyme activities of α-amylase and λ-carrageenan mixtures after PEF treatment with 0–5 cycles.

**Figure 3 foods-11-04112-f003:**
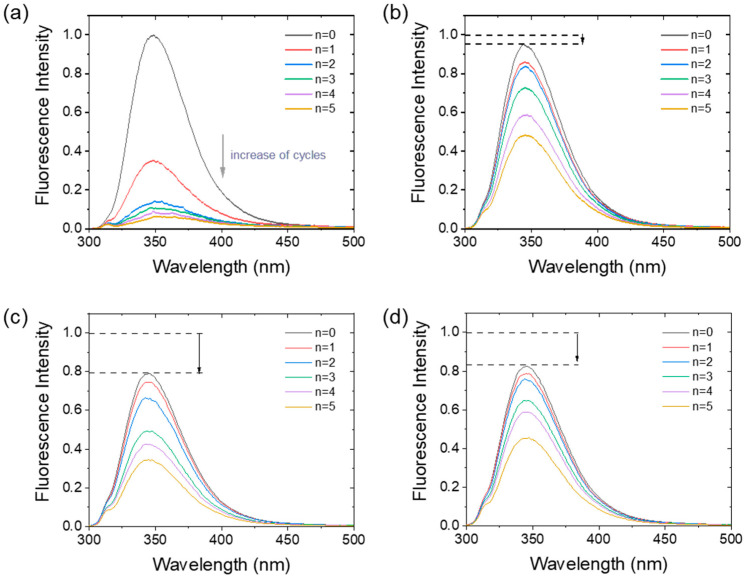
Intrinsic fluorescence of α-amylase (**a**) and α-amylase/λ-carrageenan mixtures ((**b**): 100:1, (**c**): 10:1, and (**d**): 1:1) after PEF treatment with 0–5 cycles.

**Figure 4 foods-11-04112-f004:**
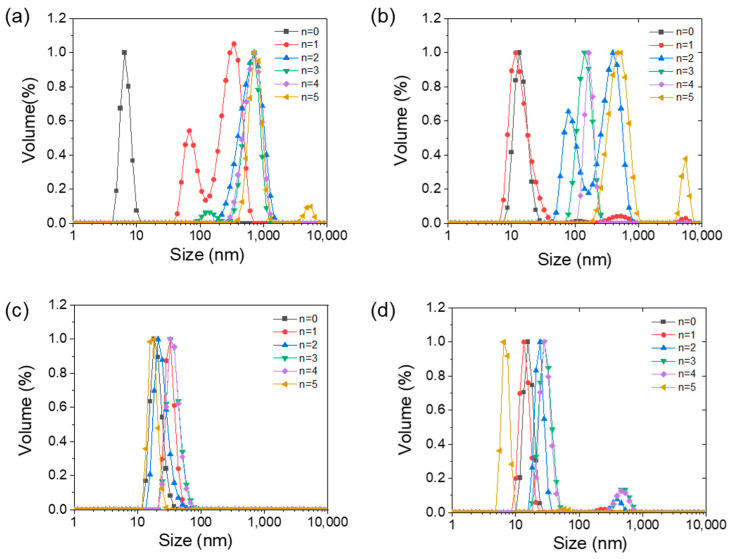
Particle size of α-amylase (**a**) and α-amylase/λ-carrageenan mixtures with a ratio of 100:1 (**b**), 10:1 (**c**), and 1:1 (**d**) after PEF treatment, n represents the number of cycles.

**Figure 5 foods-11-04112-f005:**
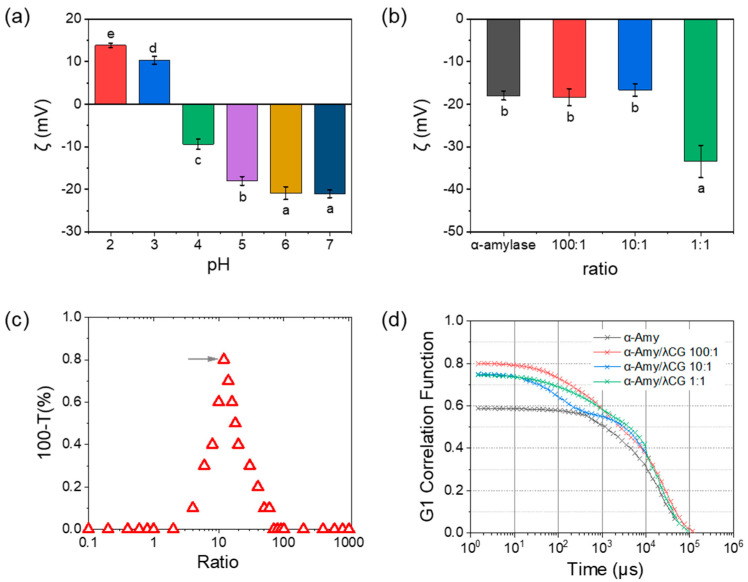
Complexation between α-amylase and λ-carrageenan, (**a**) ζ-potential of α-amylase at different pH values and (**b**) α-amylase and λ-carrageenan mixtures at pH 5.0, (**c**) turbidity of α-amylase and λ-carrageenan mixtures with various ratios, and (**d**) auto-correlation function of α-amylase and λ-carrageenan mixtures. Letters a-d inserted in figures for degree of significance, *p* < 0.05.

**Figure 6 foods-11-04112-f006:**
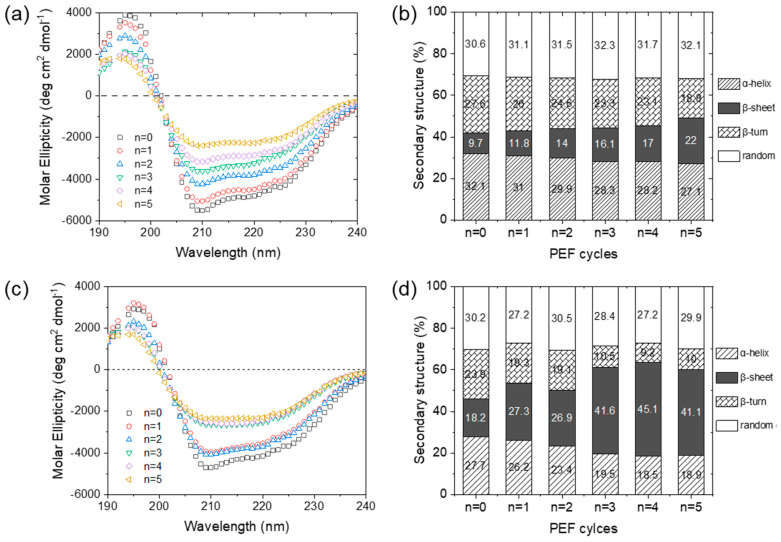
Secondary structure of α-amylase (**a**,**b**) and α-amylase/λ-carrageenan complexes with a ratio of 10:1 (**c**,**d**) after PEF treatment, n represents the number of cycles.

**Figure 7 foods-11-04112-f007:**
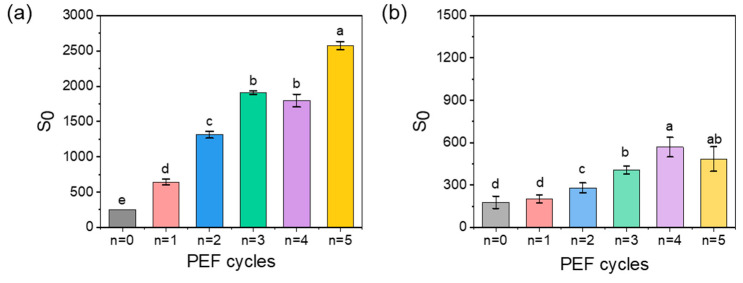
The *S*_0_ of α-amylase (**a**) and α-amylase/λ-carrageenan with a ratio of 10:1 (**b**) after PEF treatment, n represents the number of cycles. Letters a-e inserted in figures for degree of significance, *p* < 0.05.

**Figure 8 foods-11-04112-f008:**
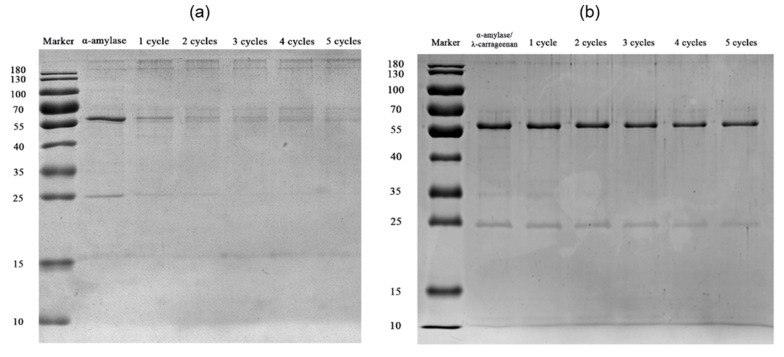
SDS-PAGE of α-amylase (**a**) and α-amylase/λ-carrageenan complexes with a ratio of 10:1 (**b**) after PEF treatment, n represents the number of cycles.

## Data Availability

The data presented in this study are available on request from the corresponding author. The data are not publicly available due to uncompleted funding project.

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
