# Peer review of "Structural Transitions of Alpha-Amylase Treated with Pulsed Electric Fields: Effect of Coexisting Carrageenan"

_foods, 2022, doi:10.3390/foods11244112_

Round 1

Reviewer 1 Report

The manuscript entitled "Structural transitions of alpha amylase treated with pulsed electric field: effect of coexisting carrageenan" is well written and presented.

The paper discusses that co-existance of certain gums can influence protein states and the protein structure can be modified by means of pulsed electric field.

The claims made by authors are adequately supported with the characterization studies. 

The present work can be used further to regulate the unfolding process of enzymes and may be used to design more studies on PEF.

Reviewer 2 Report

Reviewers’ Comments

Journal:  Foods

Paper title:  Structural transitions of alpha-amylase treated with pulsed 1 electric fields: Effect of coexisting carrageenan

Authors: Junzhu Li a , Jiayu Zhang a , Chen Li a , Wenjing Huang a , Cheng Guo a , Weiping Jin a, b* , Wangyang Shen a and b*

Comments and Suggestions for Authors

Junzhu Li and coworkers present the manuscript “Structural transitions of alpha-amylase treated with pulsed 1 electric fields: Effect of coexisting carrageenan”. This is an interesting study well conducted and concluded. I have only moderate concerns which must be fully addressed.

Introductions section

(page 1-2, line 43-44) As the authors mention in the content of the introduction, Yurij et al. investigated that λ-carrageenan increased the helix of lysozyme, made tryptophan residues expose to hydrophobic nonpolar environment, and decreased the thermal stability of lysozyme. In that case, what would be the relevance in the effect of coexisting carrageenan?

Materials and methods section 

(page 2, line 77) The α-amylase and λ-carrageenan stock solutions were mixed at the ratios 76 of 1:1, 10:1, and 100:1

Would you explain this ratios?

(page 3, line 121) Sodium dodecyl sulfate-polyacrylamide gel electrophoresis (SDS-PAGE): quantities (µg of enzymes) must be added instead of volumes

Results

Figure 2: the amylase activity expressed on % could be replaced with residual activity (or change the units).

 (page 10, line 277)SDS-PAGE of α-amylase and complexes after PEF treatment: more explanation is needed (on the bottom, the band of 25kDa)

Reviewer 3 Report

In general terms, the manuscript is well written and scientifically sound. The methodology and results are coherent.

I suggest revising the conclusions since they are very unclear. I would expect a detailed mention of the best PEF conditions related to amylase and carrageenan, based on the wide amount of analyses and results found.
